# The Extended Effect of Top-Dressed Recovered Struvite Fertiliser on Residual Irish Grassland Soil Phosphorus Levels Compared to Commercial Phosphorus Fertiliser

Ciarán O'Donnell [1], Denise Barnett [1], Joe Harrington [2] and Niamh Power [1,*]

1. Sustainable Infrastructure Research & Innovation Group, Department of Civil, Structural and Environmental Engineering, Munster Technological University, T12 P928 Cork, Ireland; c.odonnell@mycit.ie (C.O.); Denise.Barnett@mtu.ie (D.B.)
2. Sustainable Infrastructure Research & Innovation Group, School of Building & Civil Engineering, Munster Technological University, T12 P928 Cork, Ireland; Joe.Harrington@mtu.ie
* Correspondence: niamh.power@mtu.ie

**Abstract:** Phosphorus (P) is an essential growth-limiting nutrient that cannot be synthetically produced. Mineral P fertilisers are typically applied to crops to sustain modern farming practice and food production. These fertilisers are generally derived from finite phosphate ore, for which there is much concern over long-term sustainability. To address these concerns, various technologies have been developed to recover P from municipal wastewater treatment plants. One product recovered from these processes is struvite, which is a precipitate formed of magnesium, ammonium, and phosphate. To analyse the fertilisation value of the recovered struvite, field trials were conducted on perennial ryegrass (*Lolium perenne*) over three growing seasons, analysing the dry matter yield of recovered struvite fertiliser. The trial was based on a three-crop silage system designed to mimic typical Irish agricultural practice. This research highlights that recovered struvite as a fertiliser provides additional benefits including increasing the soil P levels. The struvite test case produced a statistically significantly increased soil P level from the baseline of Morgan's extractable P content of 6.4 mg/L to the optimum Morgan's soil P level of 11.13 mg/L. The findings of this research provide insight into the added benefits of recovered struvite fertiliser as a sustainable renewable P fertiliser.

**Keywords:** struvite; magnesium ammonium phosphate; recycling fertilisers; field trials; grassland; crop yield

## 1. Introduction

### 1.1. Phosphorous—An Introduction

Phosphorus (P) is a naturally occurring finite mineral, regarded as a vital building block used in the growth and reproduction of all living organisms. The industrialisation of agriculture over the past 70 years has significantly changed farming practice. This shift has led to the application of mineral P fertilisers to replace P removed by crop harvests [1]. The application of mineral fertilisers has become an integral part of modern agriculture and is vital to sustaining current farming practices and food production; in 2020, Irish agricultural used 44,393 tonnes of P fertiliser, an increase of 3% on the previous year [2–4]. Mineral P fertilisers are produced by processing P-rich ore (phosphate rock) that has formed on the earth's crust, of which reserves are finite and non-renewable [5]. The bulk of world phosphate rock reserves are located in Morocco and the Western Sahara, while the north-western region of Europe (NWE) does not have a substantial phosphorus rock supply [6]. The European Union has recognised this and added phosphate rock to its critical raw material list in 2014 [7]. In a bid to reduce reliance on phosphate rock, several technologies have been developed to recover P from various municipal wastewater streams for use as P rich fertilisers [5].

### 1.2. Recovered Phosphorous

The majority of P recovery from liquid phase waste has centred around the crystallisation of magnesium salts in the form of struvite [8]; there is up to 1475 tonnes of P potentially available for recovery from Irish municipal WWTP effluent annually [4]. Struvite is produced through a wet crystallisation process, with the resultant product having an extremely low water solubility of between 1% and 5%. This low water solubility releases the P at a slower rate from the struvite granules, creating a slow-release fertiliser, unlike commercial fertilisers [9]. The slow P release pattern of struvite supplies nutrients to the soil for extended periods to improve nutrient use efficiency [10]. This low solubility makes the prediction of fertiliser effectiveness difficult through water or citric acid determinations compared to commercial fertilisers, with soil or growth trial analysis a more favourable option [11].

Typically, high solubility commercial fertilisers tend to release P to the soil solution too quickly [8]. Once released to the soil solution, it is often rapidly immobilised or bound tightly by soil processes, thereby placing it out of reach of plants [12]. The dissolved reactive P (DRP) contained within the soil solution is often prone to losses through surface or subsurface runoff, particularly after rainfall events [13]. Once P is released to the environment through runoff, it poses a significant risk to lakes, reservoirs, streams, and upper river catchments. This risk is caused by P's unique role with carbon and nitrogen in the promotion of algal growth, which depletes dissolved oxygen levels within the water resulting in disturbance to aquatic life and potential fish kills [14]. To reduce the potential of P losses from applied mineral fertiliser, Hart et al. 2004 state that using slow release fertilisers like crushed phosphate rock could significantly decrease the potential for P loss [13]. Different research work outlines the beneficial performance of recovered struvite fertilisers in pot trial experiments, with widespread agreement on the necessity of conducting long term field trial experiments to determine the effectiveness of struvite P fertiliser for soil types and cropping systems [9,11,15,16].

For P recovery and reuse as a P-rich fertiliser to be truly successful in an Irish context, the recovered product must be comparable to mineral P in terms of crop performance and practicality of application. Struvite is noted as a root-activated fertiliser, with P release accelerated by the excretion of organic acids from the plant roots, necessitating the application of the fertiliser into the rooting zone [9,17]. This creates difficulties when applying struvite fertiliser to existing grass swards, as standard application practices apply the fertiliser to the surface (top dress). This may negatively impact the recovery and reuse of P in Ireland, with a Central Statistics Office, Crops and Livestock survey detailing that 4.5 million ha of agricultural land was used in 2018 [18], with approximately 81% (3.6 million ha) of Irish agricultural area devoted to grassland (silage, hay, and pasture) [18]. For the research to be truly applicable to Irish agriculture, the trial should be conducted on a grass-based agricultural system.

### 1.3. Focus of the Paper

The objective of this research was to conduct long duration field trials on a typical Irish agricultural system to analyse the impact of recovered struvite fertiliser on grass dry matter yields and residual soil P levels. The field trials were conducted on perennial ryegrass (*Lolium perenne* Oakpark) and compared the dry matter (DM) yield of top-dressed struvite to high solubility commercial fertiliser. The effects of struvite application on soil P levels were reviewed in a bid to determine if the recovered phosphorus product's low water solubility creates a resistance to leaching, thereby forming greater efficiency when building residual soil P indexes compared to commercially available Triple Super Phosphate (TSP).

## 2. Materials and Methods

The trials were conducted over a three-year period from 2018–2020 and were established on a three-crop perennial ryegrass (*Lolium perenne* Oakpark) silage system, which is common on Irish farms. Over the duration of the growing season, three individual crops

of grass were fertilised, grown, and harvested at six-week intervals. The grass DM yield was measured on a continuous weekly basis and at harvest following the three individual growing periods. Soil sampling was conducted prior to the commencement of the trial and after each growing session to determine the residual soil P content and the effects of fertilisation and the initial P and macronutrient content; characteristics of the field trial soil are presented in Table 1. The trial test bed is located on the Cork campus of Munster Technological University (Ireland) as shown in Figure 1, on a flat area of existing grass sward with a southerly facing aspect at an elevation of 23 m above sea level, with no over shading (E 563310.173, N 570752.400 ITM). The location has a mean annual rainfall level of 1228 mm as recorded at the nearby long-term Cork Airport Weather Station. This location is characteristic of typical rainfall patterns for the southern region of Ireland, with rainfall levels from 1000 to 1500 mm per year [19].

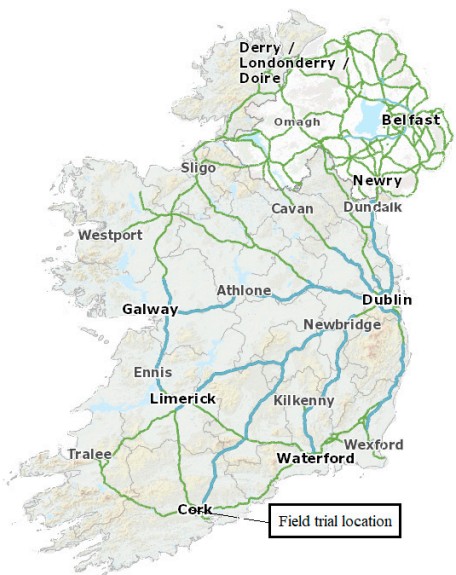

**Figure 1.** Field trial location in Ireland.

**Table 1.** Initial trial area soil properties and soil texture.

| Parameter | Unit | Method of Analysis | Results |
|---|---|---|---|
| Soil texture | - | 1 | Silt Loam |
| pH | - | 2 | 5.5 |
| Phosphorous (P) | mg/L | 3 | 6.4 |
| Potassium (K) | mg/L | 3 | 39 |
| Magnesium (mg) | mg/L | 3 | 81 |
| Calcium (Ca) | mg/L | 3 | 1090 |
| Copper (Cu) | mg/L | 4 | 2.7 |
| Manganese (Mn) | mg/L | 4 | 39.1 |
| Zinc (Zn) | mg/L | 4 | 4 |
| Boron (B) | mg/L | 4 | 0.09 |

Method of analysis, 1: sedimentation [20], 2: water [21], 3: Morgan extraction [21] (ISB, 2017), 4: DTPA [22].

The trial consisted of four test cases, tested in triplicate producing twelve 3.26 m × 6.26 m test plots stacked abreast. Each test plot contained a 5 m × 2 m growing plot (10 m$^2$) surrounded by a 632 mm wide sampling area, located along the outside perimeter of the growing plot. The sampling area was representative of the growing plot and allowed for weekly grass dry matter production measurements to be conducted using a 500 mm × 500 mm steel quadrant and shears.

The four test cases that formed the fertiliser trial are, firstly, the "Control" test case, which received no fertilisation and created the baseline comparison. The second test case was the commercial compound fertiliser and denoted as "C Comp," forming the commercial

compound benchmark of 24-2.5-10. The third test case was labelled "Eco P Comp" (Struvite plus commercial N and K to form a compound 24-2.5-10) and was designed as a direct comparison to the C Comp test case. This test case was formed by blending recovered struvite fertiliser with nitrogen and potassium to form a compound fertiliser that mirrors the C Comp test case's composition and meets the soil nutrient requirements. The fourth test case was created from the sole application of the recovered struvite and denoted as "Eco P".

*2.1. Fertilisation*

The fertiliser application rates were based on the soil sample results following Statutory Instrument 605/2017 [21]. There are numerous soil P extraction and determination methods [23]; however, national legislation adopts the Morgan's extraction process to determine soil P levels and categorises the soil P levels with the indices 1–4; it also implements the use of elemental P for fertiliser application purposes. For the purpose of this research, all P fertiliser application rates have been applied in elemental form, basing P application on soil P index of 1; this over application was conducted to review the fertiliser's ability to increase the soil index. The legislation sets the maximum quantity of P that can be spread on grassland without livestock grazing as 40 kg/ha/year for a soil P index of 1, resulting in 20 kg which is applied for the first cut, and 10 kg for each subsequent cut.

Residual grass growing over the winter period was cut and removed prior to trial commencement, using a mower to a height of 40 mm in line with Looney et al. [24]. To align the trial with fertiliser end-use conditions, the commercial compound was selected from a list of commercially available blends. An exercise was conducted to match the soil fertiliser requirements outlined in Statutory Instrument 605/2017 [21] with the commercially available fertiliser blends. It was determined that 494.2 kg of blended 24-2.5-10 (N-P-K) should be applied per hectare, with an additional 61.77 kg of muriate of potash (MOP) separately top dressed; this application forms the commercial compound test case. The selected compound fertiliser provides 118.6 kg of nitrogen, 12.3 kg of phosphorous, and 49.42 kg of potassium per hectare. The chosen compound provides 7.7 kg P less than the advised 20 kg/ha for the first cut silage. However, balanced out over the three fertilisation periods to 36.9 kg, with the annual recommended P application being 40 kg/ha.

The Eco P Compound test case was produced by this research following the guidance of the fertiliser manufacturer who produced the commercial compound. It was created by thoroughly blending struvite N and K, in the form of calcium ammonium nitrate and MOP, to create a compound fertiliser with the same ratio 24-2.5-10 of N, P, and K as the commercial compound. The trial uses crystal green struvite (magnesium ammonium phosphate), which was recovered using the Ostara Pearl recovery system from the post anaerobic digestion liquid stream of the Slough municipal wastewater treatment plant in Berkshire, England [25]. The subsequent recovered product has a typical mineral composition of 5% nitrogen, 12.5% phosphorus, and 10% magnesium, with a pearl size of $2.9 \pm 0.34$ mm pearl size [25].

All fertilisers were applied to the soil's surface (top-dressed) using a pre-calibrated 450 mm wide Einhell drop fertiliser applicator with no soil incorporation. The optimum soil pH for grassland production is at or above pH 6.3 (Wall & Plunkett, 2016) [26]. The trial location had a soil pH level of 5.5, which is below optimal. To adjust the soil pH to optimum levels, 370 kg/ha of granular limestone was applied to the trial area in accordance with the manufacturer's specification, which raised the pH to 6.1.

*2.2. Soil Sampling and Analysis*

Soil sampling was conducted in accordance with the procedure outlined in the Statutory Instrument 605/2017 [21] using a stainless steel corer to take samples to a uniform depth of 100 mm, each plot was sampled in a uniform "W" pattern covering the entire plot. At least 20 cores were taken from each plot, from which 200 g of soil cores were randomly selected to form a soil sample representative of the plot. The soil samples were forwarded

to an external INAB accredited laboratory for further analysis following the procedure set out in Statutory Instrument 605/2017 [21].

### 2.3. Grass Dry Matter Sampling

Weekly grass DM measurements were taken from the sampling belt using the shears and quadrant method. The samples were taken in a systematic pattern, ensuring that the same area was not sampled more than once in a growing season, creating a sample respective of the internal growing plot. At harvest time, the sampling belt was cut in line with the growing plot and discarded to maintain consistency between the growing plot and sampling areas. The $500 \times 500$ mm steel quadrant was positioned on the predetermined location within the sampling belt. Using electric shears, the grass within the quadrant's bounds was cut to a uniform height of 30 mm and the subsequent grass weighed. A representative sample of the grass was collected, and the DM content was determined by oven drying at 65 °C until a constant mass was achieved. From this, the DM yield was determined and presented as kg DM/ha.

The growing plot harvest was conducted six weeks after fertilisation using a reciprocating bar mower at a height of 50 mm. The internal $5 \text{ m} \times 2 \text{ m}$ growing plot was carefully marked out with the external growing belt cut first and discarded. The internal growing plot was then cut and a 300 g sample of the cut grass was collected from each plot in a Z shape pattern; this sample was bagged and labelled for further analysis to determine the crop's dry matter content. The remaining cut grass was collected and weighed to determine the fresh mass yield, following which the DM content was determined and the results presented in kilograms per hectare.

### 2.4. Statistics

The effect of the applied fertiliser on the dependant variable of grass yield was analysed using a repeated measure one-way analysis of variance (ANOVA) using the statistical software package R at a $\rho \leq 0.05$, followed by the Tukey post-hoc analys9ss. This analysis was conducted in four phases, the first of which analysed the growth response to the applied fertiliser during the inter-harvest growth period at weeks 3 and 5. The second phase of analysis reviewed the fertiliser response between treatments at each crop harvest. The third phase analysed the cumulative yearly DM yield response to fertiliser treatments and the full trial DM yield response to fertiliser treatment. The final phase analysed the residual soil P levels using ANOVA followed by Tukey post-hoc analyses.

### 2.5. Drought Damage

Following the first grass cut of the first year of the trial (2018), Ireland entered its most severe drought in 30 years [19], with precipitation levels at the Cork Airport weather station (located some 5 km from the trial site) recording an 87% reduction in mean June precipitation levels, which equated to a 37% rainfall reduction over the trial period [19]. The dry period severely impaired grass growth and allowed weeds to populate the trial area. To rectify the damage caused by the drought, the trial area was reseeded before the trial's second year. The trial area was sprayed, cultivated, and seeded with 35 kg/ha of perennial ryegrass (*Lolium perenne*, Oakpark). Rainfall patterns normalised relative to the long-term average for the remaining two years of the trial. In 2019, a 9% reduction in average rainfall patterns was recorded and in 2020, a 13% increase in rainfall was observed over the trial period [19].

## 3. Results

### 3.1. Grass Dry Matter Yield Year One

The trial commenced on the 8 April 2018, with the C Comp test case producing the most significant average quantity of DM at $7050 \pm 892$ kg/ha. This was followed by the Eco P Comp test case, which produced $6400 \pm 883$ kg/ha with no significant statistical difference occurring between the two test cases. After the first cut, the extended drought

period severely impacted grass growth, effectively stalling growth, preventing accurate weekly DM yield measurements. The second cut dry matter presented a marked decrease in DM yields across all test cases, with the C Comp test case only producing an average DM yield of $265 \pm 46$ kg/ha. The Eco P Comp test case produced $224 \pm 15$ kg/ha, followed by the $167 \pm 23$ kg and $133 \pm 32$ kg per hectare for the Control and the Eco P test case, respectively, with no statistical difference occurring between all of the second cut treatments. Precipitation levels gradually normalised in the mid to later stages of the third growth period. However, grass yields remained low in comparison to the first cut. The third cut produced a similar trend to the previous cuts, with the commercial compound test case producing the greatest mean quantity of grass DM at $2520 \pm 580$ kg/ha. The Eco P Comp fertiliser produced the second-highest DM yield of $1630 \pm 306$ kg/ha, with no statistical difference occurring between the commercial compound. This was followed by a statistically similar $350 \pm 137$ kg and $283 \pm 84$ kg of grass DM per hectare for the Eco P and the Control test cases as noted in the supplementary information presented in Table S1. The DM yield results from the first three cuts are presented in Figure 2.

The DM yield for the three first-year grass cuts were combined as a measure of cumulative annual yield and are presented in Section 3.4. The analysis highlighted no statistical difference in the dry matter yield between the commercial compound at $9839 \pm 1995$ kg DM/ha and the Eco P compound at $8304 \pm 1882$ kg DM/ha. The Eco P test case and the Control produced a comparable grass DM yield at $4586 \pm 1289$ kg/ha and $4304 \pm 1210$ kg/ha, respectively.

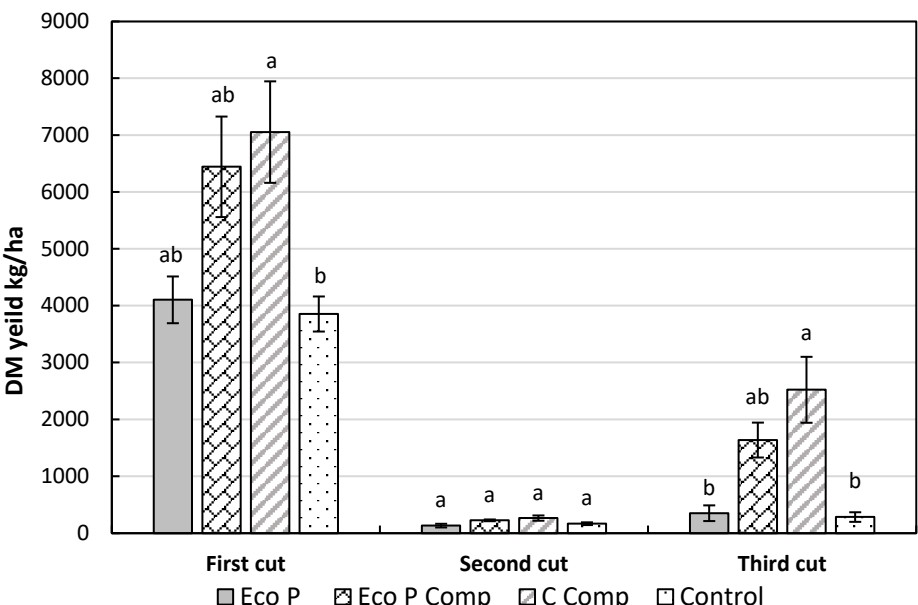

**Figure 2.** First-year effect of fertiliser treatment on grass DM yield over the first year (2018) grass harvest; within each harvest, shared letters denote that no statistically significant difference occurred ($p > 0.05$); error bars indicate standard error.

### 3.2. Grass Dry Matter Yield Year Two

The trial commenced in Year Two on 2 May 2019 following reseeding. The overall DM production of the first cut of Year Two (2019) was slightly reduced in comparison to the second and third cuts of the same year. This is a result of the sward establishment process following the reseeding that was conducted on 24 April 2019 to repair the damage caused by the previous year's drought. A persistent trend emerged through conducting weekly cuts during the three growth periods, with the C Comp test case producing the greatest weekly grass DM production relative to the Eco P Comp test case, as shown in Figure 3. The Eco P compound test case produced a statistically greater DM yield on week 5 of the second and third growth period than the Control and Eco P test case, which produced

statistically comparable DM yields throughout the three growth periods, as presented in Figure 3.

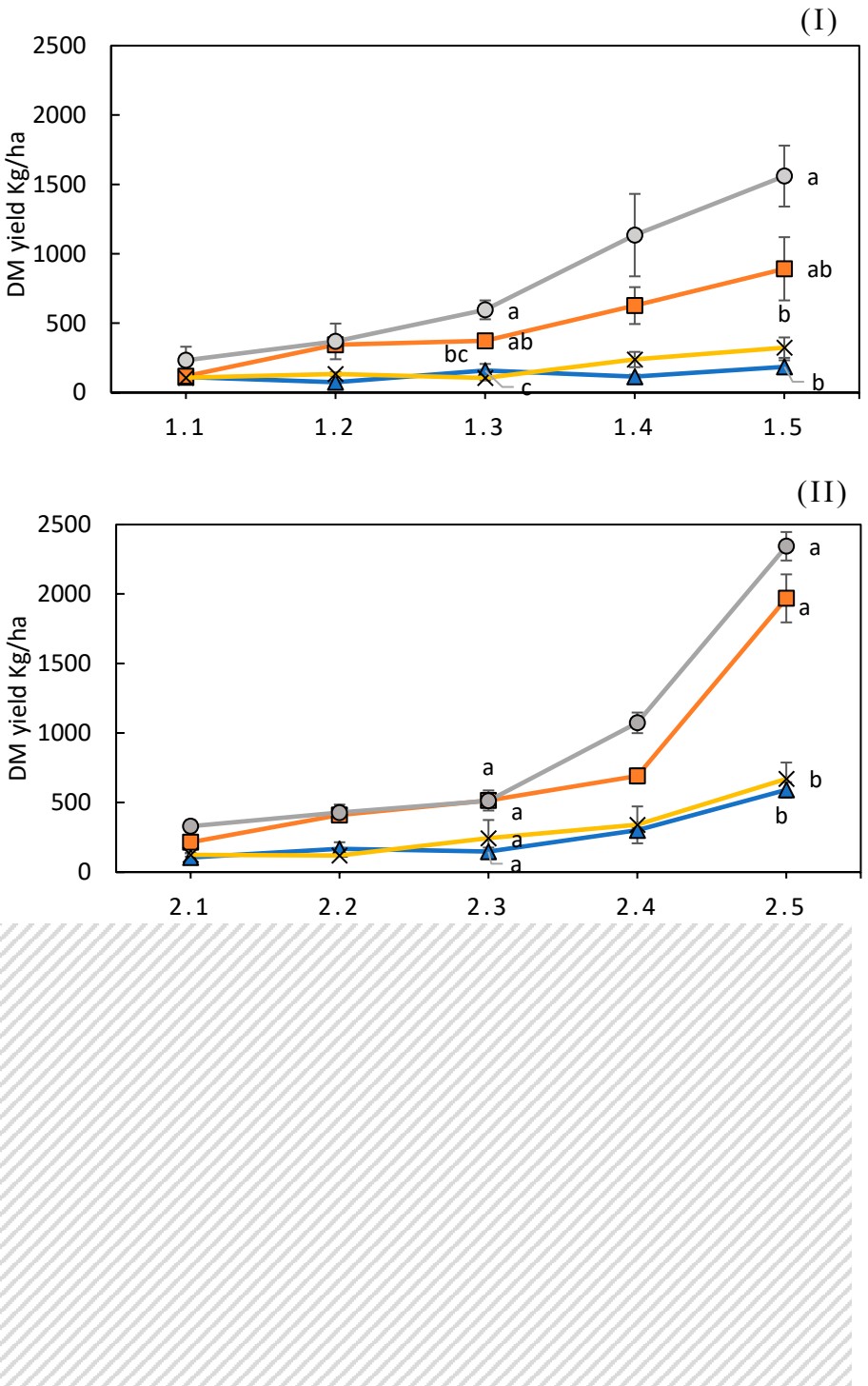

**Figure 3.** Second-year (2019) weekly growth measurements for inter harvest growth periods one to three (**I–III**). Statistical analyses are presented for weeks 3 and 5, with no statistically significant difference occurring between treatments sharing the same letter ($p > 0.05$); error bars indicate standard error.

In relation to the grass harvests the C Comp test case produced the greatest mean grass DM yield of the second-year grass cuts see Table S2, producing a statistically greater

yield then the Control and Eco P as presented in Figure 4. The Eco P Comp test case produced a statistically comparable DM yield to the C Comp test case over all individual cuts. However, when the cumulative second-year DM yield was analysed. The Eco P Comp at $7274 \pm 459$ kg/ha and C Comp at $9239 \pm 498$ kg/ha did not produce a statistically comparable yield as presented in Section 3.4.

The Eco P and the Control test case produced comparable cumulative DM yields of $2634 \pm 400$ kg/ha and $2986 \pm 452$ kg/ha. Overall, DM production increased steadily over the three cuts of the second year of the trial, in line with the sward establishment following the reseeding process.

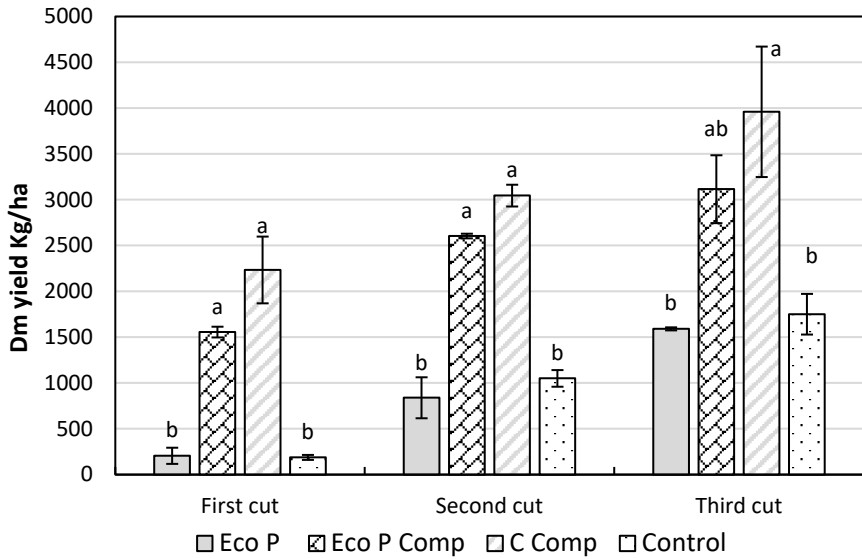

**Figure 4.** Effect of fertiliser treatment on grass DM yield over the second-year (2019) grass harvest; within each harvest, shared letters denote that no statistically significant difference occurred ($p > 0.05$); error bars indicate standard error.

### 3.3. Grass Dry Matter Yield Year Three

The third year of the field trial commenced on the 21 April 2020, under strict COVID-19 guidelines. The first growth period of the third year's growth trials produced a slight variation in grass dry matter production compared to the following growth periods, where it appears that a solid response to the applied fertiliser was produced in the early weeks of the growth period as shown in Figure 5. This may have directly resulted from the established sward and favourable environmental growth conditions in the early periods of the growing season. Overall, the C Comp test case led the growth pattern, producing the highest level of grass dry matter yield over all growing periods, a result which was statistically greater than all other test cases. In weeks one to three, the Eco P test case delivered similar growth rates to the C Comp test case, after which the C Comp test case presented an accelerated growth pattern, substantially outproducing the Eco P Comp test case until harvest.

The Eco P compound test case produced 2744, 1062, and 909 kg/ha of grass dry matter from the first cut to the third cut (see Table S3); this reduction is typical of the summer growth curve as presented in Figure 6. The results determined that the C Comp test case produced the largest cumulative yield of grass DM over the three cuts, at $9875 \pm 526$ kg/ha, which was significantly greater than all other treatments. The Eco P Comp test case produced an annual cumulative grass cut DM yield of $4899 \pm 555$ kg/ha. The Control and Eco P test cases had a statistically comparable DM yield over the three cuts. The third-year cumulative DM matter yield data is presented in Figure 7.

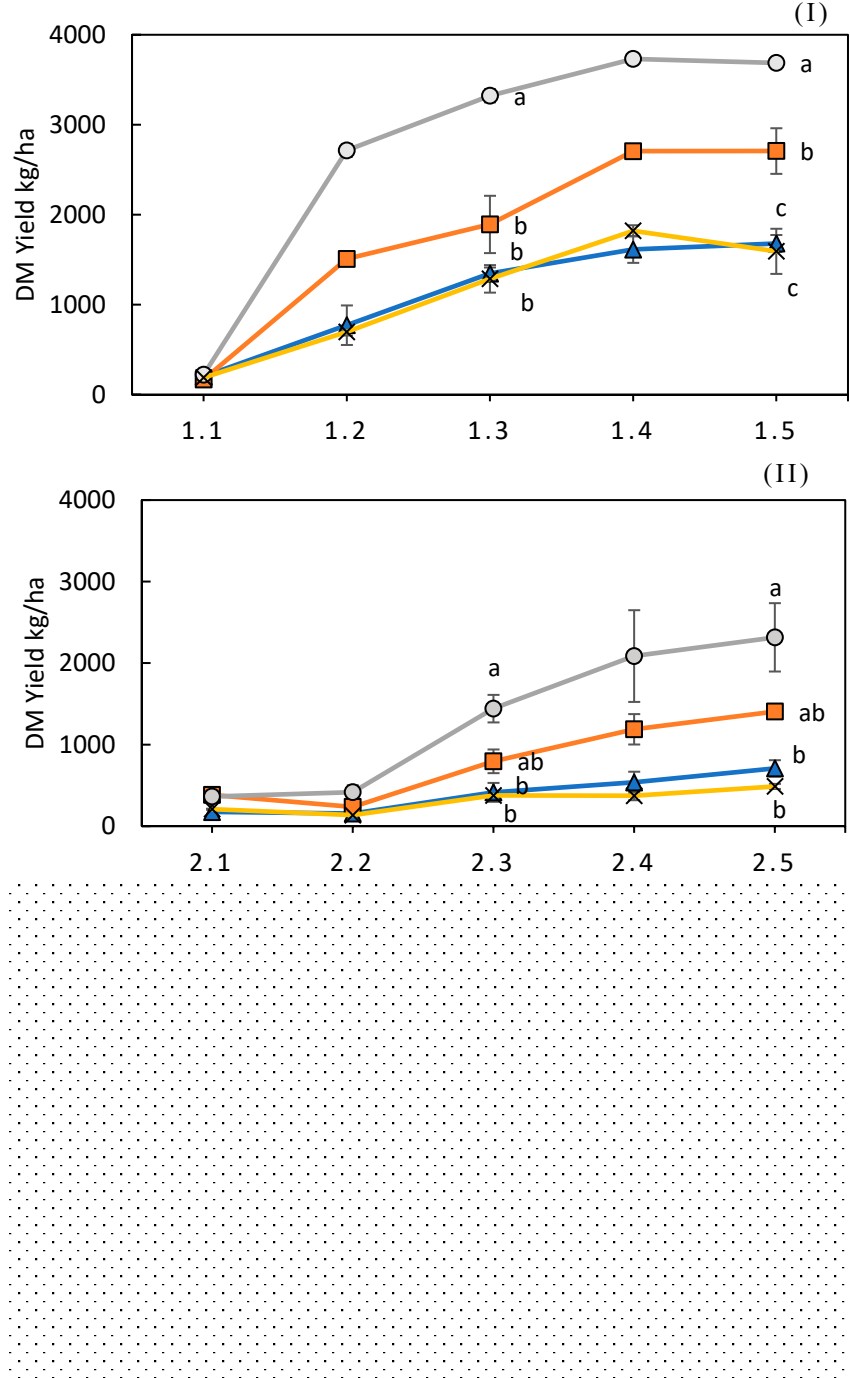

**Figure 5.** Third-year (2020) weekly growth measurements for inter harvest growth period one to three (**I–III**). Statistical analyse is presented for weeks 3 and 5, with no statistically significant difference occurring between treatment sharing the same letter ($p > 0.05$); error bars indicate standard error.

During the third year of the growth trials, investigations were conducted to determine the causes of the different DM yields between the Eco P Comp and C Comp test cases; through investigation, it was determined that the low solubility of the struvite fertiliser granules presented an issue. Throughout the trial period, the struvite granules appeared to remain intact on the soil's surface for up to 6 weeks after the initial application compared to the commercial fertilisers, which dissolved into the soil solution in a matter of days depending on weather conditions. Prior to fertiliser application on the third growing period, a random sample of struvite was collected and measured to determine the mean granule diameter. The sample size of 12 found that the struvite had a mean granule diameter of



2.9 ± 0.34 mm. At the midpoint of the third growth period, the struvite granules present on the soil surface were measured using a callipers, with a resultant mean diameter of 2.63 ± 0.39 mm.

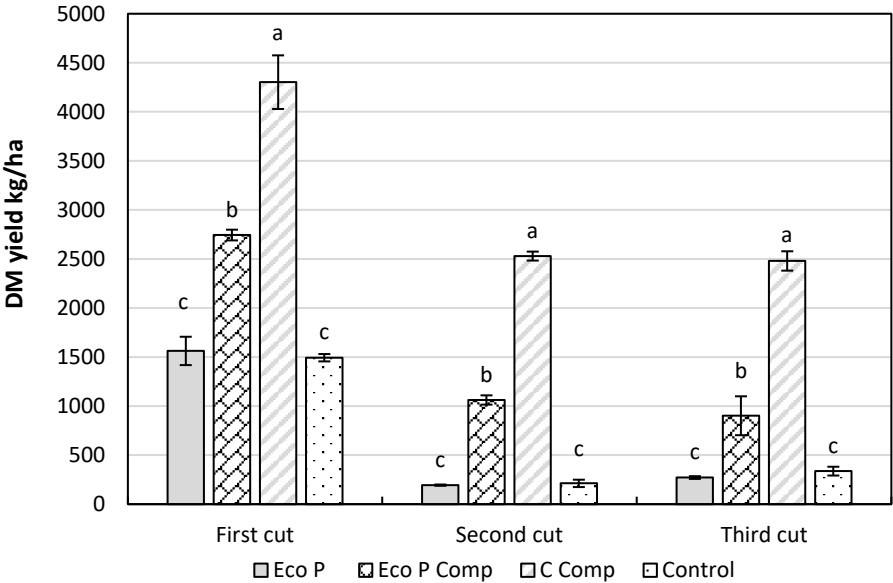

**Figure 6.** Effect of fertiliser treatment on grass DM yield over the third year (2020) grass harvest; within each harvest, shared letters denote that no statistically significant difference occurred ($p > 0.05$); error bars indicate standard error.

The measurements were repeated after the final grass cut with the struvite granules presenting a mean diameter of 1.58 ± 0.28 mm. It is clear that the degradation and breakdown of the struvite granules are essential for the release of both N and P to the plant roots; however, the release pattern is not thoroughly understood. The measurement of the struvite granules is a good indication of the slow breakdown of struvite granules but it is not an exhaustive study that tracks the P release from the struvite granules, which may be driven by numerous factors, including environmental conditions, plant type, pH, and typical soil conditions. However, it does indicate that while the struvite is in a solid form on the soil surface, the designed quantities of nutrients are not reaching the plant root during the desired growing period, affecting overall grass DM production.

### 3.4. Three Year Cumulative DM Yield

To quantify the long-term effects of fertiliser application on the grass DM yields, the results from all grass cuts over the three-year trial period from 2018 to 2020 were combined to determine the cumulative grass cut dry matter production per test case over the full duration. The grass dry matter yield followed a similar trend to the previous cuts, with the C Comp test case producing the largest cumulative quantity of grass DM at 28,954 ± 206 kg/ha over the trial duration (see Table S4). The Eco P Comp test case produced a cumulative DM yield of 20,479 ± 1008 kg/ha, which is a difference of 8475 kg/ha (34%) compared to the C Comp test case. When statistically analysed, it was determined that the C Comp test case had produced a statistically greater DM yield to all other test cases, as presented in Figure 7. The Eco P Comp test case produced a result statistically different from the other test case DM yields. The Eco P test case and the Control produced almost identical grass DM yields at 9305 ± 759 kg/ha and 9334 ± 655 kg/ha, respectively; for the Eco P test case and the Control no statistical significance occurred. These results are reflected throughout the three-year trial, with the Eco P and the Control test case producing consistently comparable grass DM quantities. This is uncharacteristic for the application of struvite fertiliser, as struvite contains 5% nitrogen. However, as noted in Table 1, the trial area has a mean Morgan's potassium (K) content of 39 mg/L, Wall and Plunkett (2020)

highlight that a soil with a Morgan's K content of less than 50 mg/L is classed as having a very low K content with a soil index of one. It is clear from reviewing the results for the three years of the field trial that the effects of struvite's N content were negated by the quantity of growth-limiting K within the soil when comparing the Control and the Eco P test cases. The cumulative DM yields over the three-year trial period are presented in Figure 7.

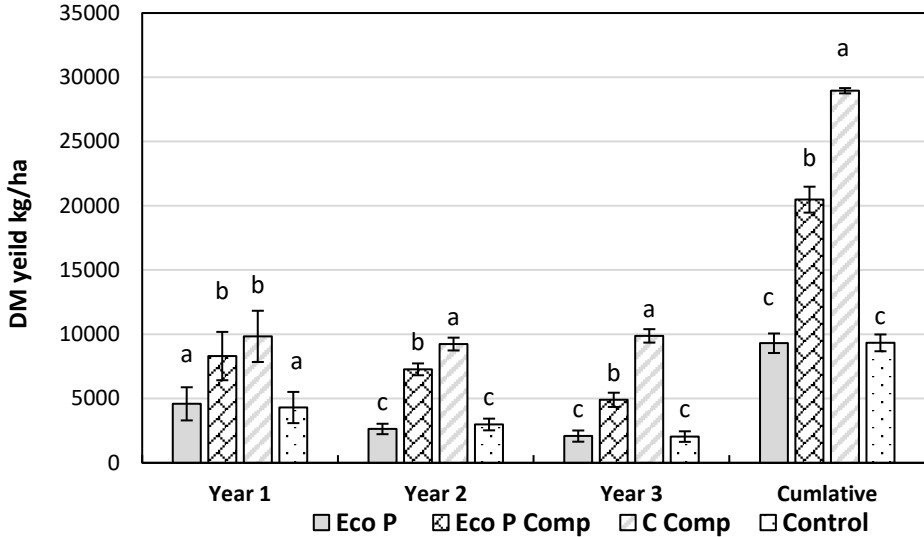

**Figure 7.** The yearly cumulative effect of fertiliser treatment on grass DM yield and cumulative yield comparison over the full trial; within each harvest shared letters denote that no statistically significant difference occurred ($p > 0.05$); error bars indicate standard error.

*3.5. Residual Soil Levels*

Baseline soil P samples were taken prior to the first fertiliser application. The samples presented a mean Morgan's soil *p*-value of 6.4 ± 0.72 mg/L, with the Control test case containing the lowest soil P content at 6 mg/L. The highest baseline soil P level was observed in the C Comp trial plots, with a mean soil P level of 6.66 ± 0.75 mg/L. These levels are in Index 3 and considered sufficient/optimum.

Over the three-year trial duration, the Morgan's soil P level of the Control test case marginally decreased by 0.17 mg/L (2.8%), resulting in an average P level of 5.83 ± 0.3 mg/L. For the C Comp trial plot, the soil P level remained relatively stable over the trial duration. It marginally by 0.56 mg/L (8.3%) over the initial mean soil P level increased following the second year of the trial, before finally decreasing to 6.53 ± 0.66 mg/L (9.68%) at the end of year three, as presented in Figure 8.

The Eco P trial plot presented the most substantial increase in residual soil P level over the period, with the soil level increasing by 4.5 mg/L above the baseline reading to 11.13 ± 0.94 mg/L. This is classified as soil P Index 4 by Statutory Instrument 605/2017 [21] and requires no additional fertilisation to maintain future grass yields. The increased soil P level directly results from receiving P fertilisation with no other macronutrients (nitrogen (N) or potassium (K)). In this case it appears that either N or K were the growth-limiting nutrients which reduced the test case biomass production, preventing P from being absorbed or removed from the soil through plant biomass production. The change in soil P levels occurred quite rapidly, in comparison to the indications from the literature that soil P build up and decline are influenced by a number of factors and can take up to eight years [26].

The Eco P Comp test case produced the second-largest increase in soil P levels of 4.43 mg/L (67%) over the baseline reading. The soil P levels increased positively after year one to 9.1 ± 0.44 mg/L (37.87%) before declining by 0.34 mg/L (19.8%) after year two, following which soil P levels increase sharply to 11.13 ± 1.4 mg/L (51%). This increase

in soil P concentration is comparable to the Eco P test case; this is an important result presenting the alternative inherent benefits of using recovered struvite fertiliser to increase residual soil P levels.

The analysis grouped the soil P results into two groups; the first group contained the Eco P and Eco P Comp test cases, which presented no statistically significant difference. The second group included the Control and the C Comp test cases, between which no statistically significant difference occurred either. A statistically significant difference occurred between the two groups, indicating that Eco P and Eco P Comp test cases produced a statistically significant increase in soil P levels.

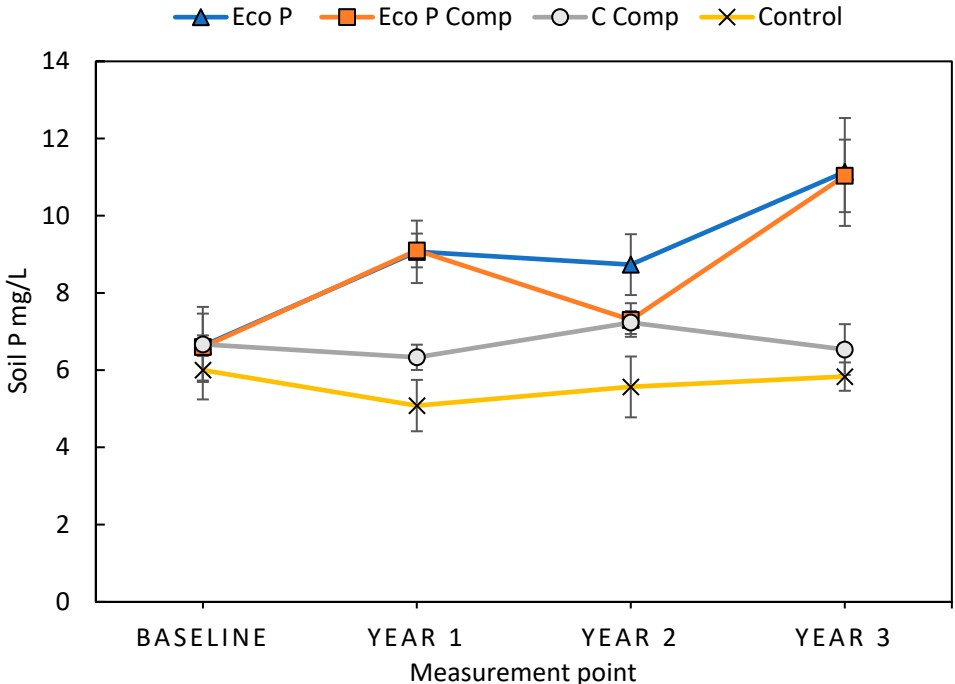

**Figure 8.** Soil P content comparing fertiliser test cases over the full duration of the field trial; error bars indicate standard error.

## 4. Discussion

Field trials pose a considerable challenge when conducting growth-dependent trials, particularly due to the impact of environmental conditions, including drought and extreme weather. Over the first year of the trial, the Eco P Comp test case produced a statistically comparable grass DM quantity to the C Comp test case. However, during the second and third years of the trial, the dry matter yield production of the Eco P Comp test case reduced. The resultant reduction in yield caused the C Comp test case to produce a statistically greater yield than all the other test cases. This may have been as a result of the struvite being applied to the soil surface (top dressed) and not incorporated into the root zone, with struvite dissolution being clearly linked to proximity to the plant roots [27]; however, typical existing fertiliser application technologies limit the application of fertiliser to surface-only application.

When the three years of the field trial data were combined to analyse the cumulative yield, a significant statistical difference occurred between the Eco P Comp and C Comp test case; the C Comp test case produced 8475 kg/ha (34%) more grass DM than the Eco P Comp test case.

The weekly grass dry matter production measurements produced a similar trend over the field trial's three-year duration. The C Comp test case produced the most notable DM yield throughout the trial, with a pronounced difference occurring after week three. This was followed by the Eco P Comp test case, which followed a similar trend at a lower yield; this lag may be a result of the slow dissolution rate of struvite, with full P release from

struvite indicated to take a time period from days to years, depending on a number of factors including struvite particle size and soil properties [15]. The Eco P and Control test case produced similar growth trends and grass DM yields throughout the trial.

The Eco P Comp test case demonstrated efficiencies in building residual soil P indexes, from the baseline of $6.4 \pm 0.72$ mg/L to $11.13 \pm 1.4$ mg/L, while producing a sustainable grass DM yield. No statistically significant increase in the C Comp test case soil P levels were observed throughout the trial. In contrast, a statistically significant difference did occur between the Eco P Comp and C Comp test cases residual soil P levels. The study outlines that Eco P fertiliser (Struvite) provides added efficiencies in building soil P indexes. This appears to be a result of the struvite's low solubility of the intact granules and consequently slower P release patterns [9]; these properties increase the competitiveness of Eco P fertilisers against commercial rock derived P fertilisers, incentivising the recovery and reuse of P from municipal wastewater streams to provide the north-western region of Europe with an indigenous supply of P with the potential to sustain modern farming practices while closing the P-cycle. The results overall are considered to be important and merit further investigation through further fieldwork; as noted by Talboys et al., there may be some advantages to applying both commercial high solubility fertilisers and low solubility Eco P fertilisers to sustain typical crop growth while building soil P indexes [9].

There is an ever growing interest in the field of P recovery and reuse, not just as a sustainable source of fertiliser, but also for environmental improvements, which P recovery can deliver, particularly from struvite, which is classed as a market ready product [5,28]. In 2021, European TSP fertiliser prices increased by a staggering 113.1% to 618 USD/tonne (EUR 531.48/tonne) [29]. It is estimated that the cost of struvite production is approximately EUR 482/tonne [30]. Accounting for P nutrient content solely, TSP is still the most financially beneficial option; however, when the values of both N and P nutrient content are considered at current European prices, struvite has a theoretical market value of EUR 728.78/tonne [29,31]. This value will promote the use of recovered P fertilisers. This value, together with tightening fertiliser supplies and increased prices, will encourage further P recovery and reuse. Large scale commercialisation of struvite fertiliser has yet to be implemented and commercial data is limited. P recovery has several benefits within the wastewater treatment process, including lowered operating costs from reduced chemical usage, lower sludge production, reduced mechanical maintenance costs, and compliance with legislation, while externally P recovery reduces the environmental impact of P emissions and creates a sustainable indigenous fertiliser [28]. Overall, it is vital that these factors are considered when estimating the cost of the recovered products and end market value in order to ensure that they can compete with commercial mineral P fertilisers.

## 5. Conclusions

Field trials were conducted over three years mimicking typical Irish agricultural practice and climatic conditions, simulating fertiliser end-use, assessing the effectiveness of recovered struvite on long term field trials, and meeting the need highlighted by Degryse et al. [15]. During each annual growing season, three individual grass crops were cut and measured to determine the DM yield and weekly grass DM measurements were taken throughout the season. The trial was conducted to analyse the DM production and residual soil P levels of recovered struvite produced from municipal wastewater treatment. This research concludes that over the three-year field trial period the recovered struvite fertiliser did not produce a statistically comparable dry matter yield to the commercial TSP fertiliser, but produced a statistical difference over the control. However, the struvite did considerably increase the soil P to above optimum levels. This is a result of struvite's inherent low solubility and may provide an advantage over traditional P fertilisers when building soil P levels while producing a sustainable yield of grass DM. Struvite recovery processes and fertilisers will need to compete with fluctuating global nutrient prices; therefore, a holistic approach must be taken when valuing recovered fertiliser to account for the additional benefits to the environmental and WWTP operations.

**Supplementary Materials:** The following are available online at https://www.mdpi.com/article/10.3390/agronomy12010008/s1, Table S1: Effect of fertiliser treatment on mean grass DM yield, over three cuts of the first year (2018) of the growth trial, Table S2: Effect of fertiliser treatment on mean grass DM yield, over three cuts of the second year (2019) of the growth trial, Table S3: Effect of fertiliser treatment on mean grass DM yield, over three cuts of the third year (2020) of the growth trial, Table S4: Yearly cumulative effect of fertiliser treatment on mean grass DM yield, and three-year cumulative grass DM yield.

**Author Contributions:** Conceptualisation, C.O. and N.P.; methodology, C.O., N.P. and D.B.; formal analysis, C.O.; investigation, C.O.; writing—original draft preparation, C.O.; writing—review and editing, N.P., J.H. and D.B.; visualisation, C.O.; supervision, N.P., J.H. and D.B.; project administration, J.H.; funding acquisition, N.P. and J.H. All authors have read and agreed to the published version of the manuscript.

**Funding:** This research has been funded by the European Regional Development Fund within the INTERREG North-West Europe VB Programme, with funding received for the Phosphorous for Your Life (Phos4you) Project.

**Institutional Review Board Statement:** Not applicable.

**Informed Consent Statement:** Not applicable.

**Data Availability Statement:** Data is available within the article and supplementary information.

**Acknowledgments:** The authors would like to acknowledge the European Regional Development Fund within the INTERREG North-West Europe VB Programme for funding received for the Phosphorous for Your Life (Phos4you) Project.

**Conflicts of Interest:** The authors declare no conflict of interest.

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
