# Peer review of "The Extended Effect of Top-Dressed Recovered Struvite Fertiliser on Residual Irish Grassland Soil Phosphorus Levels Compared to Commercial Phosphorus Fertiliser"

_agronomy, doi:10.3390/agronomy12010008_

Round 1

Reviewer 1 Report

This study presents results on rye grass response to struvite used as an alternative to traditional P fertilizers.  This is a relatively new alternative means to recycle P from wastewater and is an important and relevant topic.  The study appears to have been well carried out and merits publication.  With that said, it would have been great to test different rates of addition of the struvite.  As the authors note, the struvite is a slow release form of P and higher rates may have been more appropriate.  A follow up study should include multiple application rates of the struvite.  Secondly, the authors used the Morgan extract- it would be good for a paper so focused on P availability to use a few different soil extractions.  Mehlich III is common where I work.  I also mention that SERA 17 is a group that has published extensively on P availability and would be a good reference.  Finally some tissue analysis of the rye grass would confirm the yield and soil results and be helpful for the reader

Abstract

First two lines- P fertilizer is typically derived from mined P ore is a cleaner and more succinct way to say this

‘analyzing the dry matter production’. Was yield measured?  If not, it should be reworded to changes in soil test P

31- it is not a finite mineral, it is a mineral with finite reserves

47- how much P fertilizer was used in Ireland last year?  Give a reference here to put the quantity of struvite into perspective

60- P loss is typically through particulate erosion and much less frequently through the P in soil solution.  SERA 17 is a great source for information on soil with excess P

Table 1- values from soil tests are typically reported as mg kg rather than mg l.  Also not clear why you are reporting micronutrients, not total C or N

226- please report means +/- standard error or standard deviation

245- did you look at year as a significant factor in the ANOVA?  If year was not significant, or if the year * treatment interaction was not significant, it would be possible to present results from the three years as one figure

Also- while you mention drought in year 1- no mention is made of precipitation patterns in the following years

Did you do any analysis of grass tissue?  That would have been helpful to see if plant tissue N, P and K were similar across treatments and to confirm the low K status of the soil. 

Author Response

"Please see attachment" 

Reviewer 2 Report

The manuscript relates to the very important concept on sustainable P recycling and use in agriculture. Extensive field trials indeed are important to evaluate the efficiency of "novel" P fertilizers and the study contributes to the available data base on the subject.

The manuscript is in general well written. The language is good although some small edits might be need. In some cases the future tense is used instead of an appropriate past form e.g.

Line 88. will should be replaced by "have been" or similar

The introduction is well written but lacks a clear hypothesis. The objectives are quite vague and should be rephrased to mirror the rationale of the study clearer and in more depth.

The materials and method section is very long and many of the details presented could be omitted to reduce the length. Authors may consider substantially reducing this section.

Line 96. The ryegrass species used should be mentioned here

Line 137. See comment from Line 88

Figure 1 and all figures, please indicate what do error bars represent (stand. deviation, error?)

Line 214. What type of post hoc test was used to compare treatments?

The results are clearly presented and again the section's length should be reduced. Authors may consider to focus only on the presentation of significant results.

The discussion section could be improved. There is no discussion on the reasons leading to higher residual P level in some treatments. Nevertheless, the abstract of the manuscript highlights this as the study's main outcome. Do the authors find any potential implications on P losses from the tested treatments.

A small paragraph discussing the costs differences between P products might be also relevant to be included. If the cost of recovered P is substantially higher than mineral P, then farmers would not adopt the new products.

Author Response

"Please see document attached"
